# Clinical and Virological Characteristics and Prognostic Factors in Viral Necrotizing Retinitis

**DOI:** 10.3390/jpm12111785

**Published:** 2022-10-29

**Authors:** Léa Fitoussi, Amandine Baptiste, Adam Mainguy, Anne-Sophie L’Honneur, Magdalena Bojanova, Agnès Dechartres, Flore Rozenberg, Bahram Bodaghi, Sara Touhami

**Affiliations:** 1Department of Ophthalmology, Pitié Salpêtrière University Hospital, Sorbonne Université, 75013 Paris, France; 2Department of Public Health, Centre de Pharmacoépidémiologie de l’AP-HP (Cephepi), Pitié Salpêtrière University Hospital, Sorbonne Université, 75013 Paris, France; 3Institut Pierre Louis d’Épidémiologie et de Santé Publique, Institut National de la Santé et de la Recherche Médicale (INSERM), Pitié Salpêtrière University Hospital, Sorbonne Université, 75013 Paris, France; 4Department of Virology, Cochin University Hospital, Université Paris Descartes, 75014 Paris, France

**Keywords:** viral necrotizing retinitis, VZV, CMV, HSV, retinal detachment, outcome, viral load, prognosis

## Abstract

Purpose: Describe the clinical and virological characteristics of viral necrotizing retinitis (VNR) and assess its prognostic factors. Methods: Retrospective study (Pitié Salpêtrière Hospital, Paris) of consecutive VNR patients diagnosed and monitored by qPCR on aqueous humor between 2015 and 2019. All patients received induction therapy with intravenous +/− intravitreal injections (IVI) of antivirals. Results: Forty-one eyes of 37 patients with a mean age of 56 years were included. Involved viruses were VZV (44%), CMV (37%) and HSV2 (19%). Acute retinal necrosis represented 51%, progressive outer retinal necrosis 12% and CMV retinitis 37% of eyes. Forty-six percent of patients were immunocompromised. Median BCVA was 0.7 LogMAR at baseline and 0.8 LogMAR after an average of 14.1 months. VNR bilateralized in 27% of cases after 32 months. Retinal detachment (RD) occurred in 27% of cases after a mean duration of 98 days. Factors associated with a “poor BCVA” at 1 month were: advanced age, low baseline BCVA, high vitritis grade and viral load (VL) at baseline and the “slow responder” status (i.e., VL decrease <50% after 2 weeks of treatment). Factors associated with RD were: advanced age, immunocompetence, low baseline BCVA, high vitritis grade at baseline and use of ≤5 IVIs. Conclusions: Clinical factors including advanced age, immunocompetence, low BCVA and high vitritis grade at baseline were associated with a poor prognosis. New virological factors were predictive of a poor outcome: high baseline VL and the “slow responder” status. Sequential intraocular fluid sampling might help prognosticate the outcomes of VNR.

## 1. Introduction

VNR necrotizing retinitis (VNR) is a sight-threatening condition caused by herpes simplex virus (HSV) 1 and 2, Varicella-zoster virus (VZV), and cytomegalovirus (CMV). They include acute retinal necrosis (ARN), progressive outer retinal necrosis (PORN), and CMV retinitis. Although the diagnosis relies on medical history, immune-system characteristics and fundus features, virological identification remains important to propose the most optimal therapeutic strategy [1,2]. Reliable and rapid identification of the causal agent is enabled by real-time quantitative analysis of ocular fluids by polymerase chain reaction (PCR), which is highly sensitive and specific [1,2]. While ophthalmologists are aware of the sight-threatening complications of VNR, only very few studies have assessed their prognostic factors or examined their virological response over time. The objective was to describe the clinical and virological characteristics of VNR and evaluate the factors associated with a poor visual prognosis.

## 2. Materials and Methods

### 2.1. Study Design and Regulatory Information

This was a retrospective single-center cohort study. Institutional review board approvals for retrospective chart reviews were obtained commensurate with the respective institutional requirements prior to the beginning of the study. Described research was approved by the Ethics Committee of the French Society of Ophthalmology (IRB number 00008855) and adhered to the tenets of the Declaration of Helsinki. Fully informed consent was obtained from all patients.

### 2.2. Population

We included all consecutive adult patients with virologically q-PCR-confirmed VNR (ARN, PORN and CMV retinitis), seen at Pitié Salpêtrière University Hospital (Paris, France) between November 2015 and July 2019. All patients underwent an exhaustive work-up to exclude other causes of uveitis, including clinical examination by an internal medicine specialist, Quantiferon Gold/Mantoux reaction, TPHA-VDRL, serum antibodies against toxoplasmosis and human immune-deficiency virus (HIV), chest X-ray, serum angiotensin converting enzyme levels and other targeted tests when deemed necessary.

### 2.3. Demographic and Clinical Data

Demographic data included age, gender, medical history, previous eye surgeries and current or previous treatments. At each visit (inclusion, Month (M) 1, 3, 6, 12 and the last available visit), the following clinical information was collected: clinical symptoms, best-corrected visual acuity (BCVA), intraocular pressure, slit-lamp characteristics, grade of anterior chamber inflammation based on the SUN criteria [3] and fundus findings, including the number of quadrants affected by VNR, vitritis grade according to the Nussenblatt classification [4] and occurrence of retinal detachment (RD) during the follow-up. BCVA was assessed on a decimal scale, converted to the logarithm of the minimum angle of resolution (LogMAR). For BCVA below 20/400, the following conversions were used: 20/1000 = 1.7 LogMAR, 20/1600 = 1.9 LogMAR, 20/2000 = 2 LogMAR, 20/4000 = 2.1 LogMAR, no light perception = 2.6 LogMAR.

The therapeutic procedures used for each patient (including intravitreal injections (IVI), systemic antivirals, and/or local or systemic corticosteroids) were described.

### 2.4. Virological Assessment

Patients underwent anterior chamber paracentesis at baseline (to confirm the diagnosis) and at least once again during the follow-up for viral load monitoring under treatment (i.e., total of at least two paracenteses). Viral DNA was extracted (QiaSymphony, Qiagen, Germantown, MD, USA) from aqueous humor (AH) samples (40 μL resolved in 80 μL of eluant), as described by the manufacturer’s instructions. Real-time qPCR was performed on 20 μL of eluant corresponding to 10 μL of AH, using quantification kits (for HSV-1, HSV-2, VZV and CMV, Real-Star kits, Altona Diagnostics, Hamburg, Germany) with threshold detection and quantification levels of 500 DNA copy/milliliter for HSV-1 and HSV-2, 100 DNA copy/milliliter for VZV, and 334 DNA copy/milliliter for CMV. Results were expressed in DNA copy/milliliter (mL) or Log IU (International Unit)/mL.

### 2.5. Definition of Slow versus Rapid Responders

VNR eyes were divided into two categories depending on viral load evolution under induction therapy: rapid responders were defined by a 50% or greater decrease of their viral load (versus baseline) after 2 +/− 1 weeks of antiviral treatment. Slow responders were defined by a viral load decrease of less than 50% after 2 +/− 1 week of this treatment.

### 2.6. Outcome

Poor visual outcome was defined by either one of the two following surrogates:-Poor BCVA ≤ 20/100 decimals (i.e., ≥0.7 logMAR) after one month (M1) of treatment.-Occurrence of RD during the follow-up.

### 2.7. Statistical Analyses

Statistical analyses were carried out using the R software (version 3.6.3, https://cran.r-project.org/). Performed tests were two-sided and used a significance level of 5%. Because of the small number of patients with bilateral follow-up, we did not take into account the correlation of eyes in bilaterally affected patients. Qualitative factors were reported with frequencies and percentages. Quantitative variables were described with means (+/−standard deviation) or medians [1st quartile–3rd quartile]. For the analysis of factors associated with a poor visual outcome, proportions were compared using Pearson Chi2 tests (or Fisher exact tests if needed), and comparisons of quantitative variables were performed using non-parametric tests (Wilcoxon rank test or Kruskal-Wallis test). For the variables most significantly associated with the two assessed deficits (low BCVA or occurrence of RD), a univariate and multivariate logistic regression model was used to estimate their effect size. Odds ratios and their corresponding 95% confidence intervals were reported, and a likelihood ratio test was performed.

## 3. Results

### 3.1. Characteristics at Baseline

We evaluated 41 eyes from 37 patients (14 females) (Table 1). Fifty-one percent of eyes (21/41) had ARN, 12% (5/41) had PORN and 37% (15/41) had CMV retinitis (Appendix A). Quantitative PCR on aqueous humor detected HSV1 in 0%, HSV2 in 19% (8/41), CMV in 37% (15/41), and VZV in 44% (18/41) of eyes (Table 1).

The mean age at diagnosis (“baseline”) was 56.7 years (range: 24–90), significantly lower in patients with HSV2 than those with CMV and VZV necrotizing retinitis (*p* = 0.005). Seventeen of 37 patients (45.9%) were immunocompromised, including 100% of CMV retinitis patients and 25% (4/16) of those with VZV. All HSV2-infected eyes were diagnosed in immunocompetent patients (Table 1). Among the 17 immunodepressed patients, 4 (23.5%) were HIV-positive (3 with CMV retinitis and 1 with VZV-PORN). The baseline characteristics of immunosuppressed individuals, depending on the cause of immunodepression and causal virus, are displayed in Appendix A.

At diagnosis, 5/41 patients (12.2%) had bilateral disease. The median baseline BCVA was 0.7 LogMAR [IQR: 0.3–1.7] (approximately 20/200), slightly but nonsignificantly lower in VZV versus CMV and HSV2 infected eyes (Table 1). The median grade of vitritis was 2+ cells [IQR: 1.2–3]. Vitritis was ≤2+ in 62% (24/39) of eyes and >2+ in the remaining 38% (15/39) of cases. The median number of retinal quadrants affected by necrosis was 1.2 [IQ: 1–4], and 35% (14/40) of eyes had >2 quadrants involved (Table 1).

### 3.2. Follow-Up and Treatment

Patients were followed up for a mean period of 14.1 +/− 13.6 months (range: 1–39 months) with a median number of six visits [IQR: 4–9]. The mean time from the first symptoms to treatment initiation was 21.7 +/− 44.4 days. It was longer in CMV (54.2 days) versus HSV2 (4.9 days) and VZV eyes (10.2 days) (Table 2). It was also longer in immunodepressed (38.5 days) versus immunocompetent (9.1 days) patients (*p* = 0.0032, Appendix A). The mean duration of intravenous (IV) treatment was 23.5 days. Fifty-four percent of patients (20/37) received IV aciclovir, 19% (7/37) received IV ganciclovir and 60% (22/37) received IV foscarnet, meaning that eight patients were switched from one molecule to another to match the viral diagnosis once it was made (Table 2). Maintenance therapy consisted of oral valganciclovir in 32% (11/34) of patients (11 CMV retinitis), oral valaciclovir in 65% (22/34) of patients (15 VZV and 7 HSV2 retinitis) and oral famciclovir in 3% (1/34) of patients (1 VZV). Thirty-five of 41 eyes (85.3%) received at least one IVI of ganciclovir. The mean number of ganciclovir IVIs was 6.8 +/−7.7. Eleven of 41 eyes (26.8%) received at least one foscarnet IVI. Seven eyes (17.1%–7/41) received ganciclovir then foscarnet IVIs. Only 4.8% (2/41) of the studied eyes did not receive IVIs (both eyes had CMV retinitis) (Table 2). None of the patients received oral or IV steroids during the follow-up.

### 3.3. Baseline Viral Load and Its Evolution Overtime

All patients had viral load quantification by qPCR at baseline, and at least one additional time during the study period. The mean number of viral load quantifications during the follow-up was 7.8 +/− 5.8. At baseline, the mean viral load (VL) was 6.6 +/− 1.4 Log10 IU/mL. It was higher in VZV (7.4 +/− 1.0 Log10 IU/mL) versus CMV- (6.2 +/− 1.1 Log10 IU/mL) and HSV2-infected eyes (5.3 +/− 1.5 Log10 IU/mL), *p* = 0.001 (Table 1). The raw VL figures were higher in both immunodepressed and immunocompetent VZV-infected eyes when compared to CMV (all of which were immunodepressed) and HSV2 (all of which were immunocompetent)-infected eyes (Appendix A). In other words, the VL was higher in VZV-infected eyes independently of the immune status. Regarding VL evolution over time, there were 52.8% (19/36) of rapid responders (Table 2): 54% (7/13) in eyes with CMV retinitis, 67% (4/6) in eyes with HSV2 retinitis and 47% (8/17) in those with VZV retinitis. Although numerically lower in the subgroup of immunocompetent VZV-infected eyes, the rate of rapid responders did not statistically differ depending on the virus (*p* = 0.84) or immune status (*p* = 0.74, Appendix A). The kinetics of VL decrease during the first 2 +/− 1 weeks of treatment is shown in Figure 1. The curves representing VL kinetics for each eye according to the causal virus showed a clear tendency to decrease (from week 0 to week 3) in rapid responders (independent of the virus type), as opposed to slow responders whose VL seemed to remain stable or increase. 

### 3.4. Visual Outcome

The mean baseline BCVA was 0.9 +/− 0.8 LogMAR for CMV, 0.8 +/− 0.7 LogMAR for HSV2 and 1.1 +/− 0.8 LogMAR for VZV (*p* > 0.05) (Table 1). Figure 2 and Appendix A show that BCVA decreases between baseline and M3 in the CMV (mean difference between M3 and baseline: +0.2 +/− 0.8 LogMAR) and VZV group (mean difference between M3 and baseline: +0.5 +/− 1.1 LogMAR) and not in the HSV2 group in which BCVA shows stability (mean difference between M3 and baseline: −0.02 +/− 0.7 LogMAR). On the other hand, BCVA evolution between the 3rd and 12th month of follow-up, shows improvement in the CMV group, stabilization in the HSV2 group and worsening in the VZV group. Immunocompetent patients with VZV infection seemed to display the lowest BCVA figures during the follow-up (Appendix A).

### 3.5. Bilateral Disease

Bilateralization of VNR occurred in 27% (10/37) of patients after a mean duration of 32 +/− 41.8 months (range: 0–60) (Table 2).

In immunosuppressed patients (N = 17), six patients developed bilateral disease (5/13 with CMV retinitis and 1/4 with VZV retinitis) (Appendix A). Among the six patients with bilateral disease, four (including 3/5 in the CMV group and 1/1 in the VZV group) had bilateral disease at presentation. Only two patients (both of which had CMV) developed bilateral disease during the follow-up: one did seven months after initial treatment (i.e., under low doses of oral antivirals) and the other almost one year after the first episode (i.e., after discontinuation of treatment).

In immunocompetent individuals (N = 20), four patients developed bilateral disease (3/8 in the HSV2 group and 1/12 in the VZV group). Three of four patients bilateralized after discontinuation of antivirals (i.e., >1.5 years of initial diagnosis) and one did after one month of treatment because therapy was lowered too quickly. Therefore, 50% (N = 5) of patients with bilateral disease had bilateral involvement at or close to diagnosis. In other cases, bilateralization occurred at decrease or withdrawal of antivirals.

### 3.6. Retinal Detachment

RD occurred in 11/41 eyes (26.8%) after a mean follow-up of 99 +/− 123.9 days (Table 2). RD occurred more frequently in immunocompetent (38.1%) than in immunodepressed (15%) patients (*p* = 0.09, Appendix A). Among immunodepressed patients, RD occurred in only three eyes during the follow-up, all of which were infected by CMV. The cause of immunodepression in these patients was not HIV (Appendix A). BCVA evolution according to the presence or absence of RD during the follow-up is presented in Appendix A. Overall, retinal detachment was associated with lower BCVA figures over time when compared to the absence of RD (statistical significance reached at baseline, Month 1 and 12).

### 3.7. Prognostic Factors of Poor Visual Outcome

#### 3.7.1. Factors Associated with Poor BCVA at One Month of Treatment Initiation

Sixty-three percent of eyes had a poor BCVA at M1. Factors associated with this surrogate outcome were low baseline BCVA, older age at diagnosis, high vitritis grade and viral load at baseline and the “slow responder” status (Table 3 and Table 4). These latter two virological factors were correlated with poor prognosis on both uni- and multivariate analyses (Table 4). A high number of retinal quadrants affected by VNR was close to being significantly associated with a poor visual outcome (*p* = 0.07, Table 3). On the other hand, causal virus and presence of immunodepression were not statistically correlated with the fact of having BCVA ≤ 20/100 at month 1 (Table 4). Interestingly, the combination of immunocompetence and VZV infection seemed to correlate with the highest likelihood of having a poor BCVA at month 1 (i.e., in 92.3% of cases, Appendix A). In immunodepressed patients, the cause of immunodepression (HIV versus other causes) did not play a role in predicting this outcome (Appendix A).

#### 3.7.2. Factors Associated with the Occurrence of Retinal Detachment during the Follow-Up

Twenty-seven percent (11/41) of eyes developed RD during the follow-up after a mean period of 98 days (range: 0–330). Factors associated with this surrogate outcome were low baseline BCVA, high vitritis grade at baseline and a number of antiviral IVIs < or =5 (Table 5, Appendix A). Older age and immunocompetence were close to be significantly associated with an increased likelihood of developing RD.

#### 3.7.3. Role of Immune Status

The role of immune status is investigated in Appendix A. Of 37 patients, 17 were considered immune compromised, of whom 13 had CMV retinitis and 4 had VZV retinitis. The causes of immunodepression were the following: ongoing chemotherapy for cancer (N = 5), ongoing immunosuppressive treatments for graft versus host disease (N = 2), ongoing immunosuppressive treatments for autoimmune disease (N = 2), ongoing immunosuppressive treatments for organ transplant (N = 4) and HIV (N = 4). Their distribution was similar among CMV retinitis and VZV-infected patients (*p* = 0.58). Of note, only 4 patients (5 eyes: 3 with CMV and 2 with VZV) had HIV, which shows that CMV retinitis in this cohort cannot be considered as an AIDS defining disease. The baseline viral load was not higher in immunodepressed versus immunocompetent patients, nor was it higher in patients with HIV versus those with other causes of immunodepression (Appendix A). The presence of an immunodepression was not predictive of the occurrence of a BCVA ≤ 20/100 at month 1 (*p* = 0.24), retinal detachment (*p* = 0.095), nor was it associated with a higher likelihood of being a rapid or slow responder (*p* = 0.74) (Appendix A). Appendix A shows that HIV does not confer a different outcome (in terms of BCVA ≤ 20/100 at month 1, occurrence of RD or rate of rapid responders, all *p* > 0.05) when compared to the other causes of immunodepression.

Regarding the interaction between the immune status and causative virus, it seemed that immunocompetent patients with VZV retinitis were the likeliest to have a poor BCVA outcome at month 1 or develop RD overtime (Appendix A). Their baseline viral load was higher (Appendix A).

## 4. Discussion

Retinal necrosis is a group of disorders of the posterior segment, induced mostly by viruses of the herpes virus family but also by other pathogens (e.g., *Toxoplasma gondii*) or non-infectious diseases like Behçet’s disease [5,6]. These rare yet serious conditions can be potentially blinding [5,6]. Molecular diagnosis on aqueous humor samples is commonly used to identify the culprit virus [5,6]. While previous publications analyzed the monitoring of VNR by qPCR [7,8,9,10,11], only very few studies looked at the virological predictive factors of visual outcome. To our knowledge, this is the largest study investigating the follow-up of VNR and the clinical and virological predictive factors of visual outcome.

Regarding the demographic characteristics, our cohort includes mostly middle-aged men, which is comparable to the literature [7,8,9]. HSV2 necrotizing retinitis occurs in younger patients than VZV or CMV VNR, in agreement with our findings [12]. The immunosuppressed status found in all patients with CMV retinitis was congruent with the literature [9,11] as were the baseline visual acuity [9,13,14,15,16,17,18,19], baseline vitritis grade [9] and number of retinal quadrants initially involved [9,13,14,20,21]. Immunodepressed patients with CMV retinitis were rarely affected by HIV (only 4 of 13 patients), which shows that CMV retinitis in this cohort cannot be considered as an AIDS-defining disease.

Recent studies published RD rates around 50% [7,8,20,22,23,24] following VNR caused by VZV and HSV, which differs from our 27% rate. This might be explained by our aggressive therapy consisting of a mandatory IV induction treatment (of 2–3 weeks) followed by an oral maintenance therapy (minimum of 6 months), almost systematically associated with intravitreous injections (90% of eyes). Another explanation might be related to the time of evaluation: 50% of patients were lost to follow-up at month 12, therefore, longer follow-ups may yield higher RD rates. Finally, cited studies included only VZV and HSV VRN; however, our RD rate remains lower even when we exclude CMV infected eyes.

We found higher initial VLs than in other studies [7,9], particularly for CMV [11]. This could be explained by differences in evaluation time, qPCR tests, genetic background or differences in viral subtypes as they may differ depending on geographic location. Only a few studies reported on VL evolution in VNR. Our patients showed either a rapid or slow VL decline without going through a flattened phase in contrast to what Bernheim [7] and Hafidi and al. published [9] (data not shown). Our study also potentially highlights two new types of patients: rapid responders, presenting a decrease in their VL > or =50% after 2 +/− 1 weeks of treatment initiation; and slow responders in the opposite situation. An explanation for this difference could be that our qPCR timepoints may be different from those previously described. Interestingly, we noticed that the VL was still measurable for a small yet significant proportion of eyes (7%), after 5 months of follow-up. This class of “very slow responders”, should be the subject of further studies. Calvo and al. [8] and Bernheim et al. [7] had already noticed persistently positive high VLs raising questions about the ideal duration of antiviral therapy. The average duration of IV antiviral therapy was 23.5 days, close to that of Hafidi et al. [9] who indicated that they stopped intravenous antivirals once they obtained clinical proof of healing (e.g., retinal pigmentation).

Regarding the predictive factors of poor visual evolution, we analyzed them by focusing on two surrogate outcomes: occurrence of RD during the follow-up and “low BCVA” at one month of antiviral initiation. Although arbitrary, the choice of such surrogates seemed clinically relevant as it has been shown that RD occurrence was associated with poor visual outcomes in VNR [14,25,26] and other types of uveitis [27] but also with increased medico-economic burden. Furthermore, the choice of the M1 BCVA outcome is explained by the fact that it was found positively correlated with BCVA figures at longer follow-up times in our cohort (6 and 12 months, Spearman correlation coefficients of 0.88 and 0.89) (data not shown).

Regarding the surrogate outcome “RD occurrence”, we found that it occurred more frequently in older patients. This surprising finding can be explained by the fact that older patients may be more prone to retinal damage, possibly due to weakened immune responses. A high vitritis grade was found to be associated with RD occurrence. An explanation to this may be that inflammatory vitreous changes might induce abnormal peripheral tractions and retinal tears. The causal virus was not associated with RD occurrence, although VZV was relatively over-represented (54.5% of patients with RD), consistent with previous literature [28,29]. Although controversial, we found a significantly lower RD rate in eyes receiving more than five IVIs, in line with other papers [9,14,30]. It is also relevant to note that VL decline, whether rapid or slow, did not influence the occurrence of RD.

Regarding the surrogate outcome “poor BCVA at M1”, we found that low baseline BCVA, high baseline vitritis grade and larger retinal necrotic areas (close to statistical significance) were predictive of a poor outcome. The novelty of this paper relies on the virological data showing that eyes with a low baseline viral load and rapid responders (in terms of VL kinetics at 2 +/− 1 weeks) have significantly better BCVA at one month. Such an association between VL kinetics and the visual outcome has never been reported so far and is of interest as it can help physicians foresee the visual evolution of their patients. In multivariate analysis, the prognostic value of the baseline VL and slow or rapid responder status was shown to be independent of the causative virus and immune status. This shows the strength of our viral parameters as predictors of visual outcomes in VNR. Reasonable implications of this would be to propose systematic sequential intraocular fluid sampling to all patients with VNR (at least during the first three weeks of treatment). In slow responders, escalating therapy might be proposed, including more frequent intravitreal injections or use of different antivirals and testing for resistance when the clinical situation does not improve [15]. In this study, we however could not find any clear cut-off in viral load that could be used to choose the most appropriate timing for the switch to prophylactic antiviral doses (data not shown).

It is interesting to note that although the prognostic value of the aforementioned presented virological parameters was mathematically independent of the causative virus and immune status, this study still points toward a specific category of patients that seems to show the worst prognosis, namely immunocompetent patients with VZV infection. Such patients display the highest baseline viral loads, lowest rates of rapid responders and worst anatomic and functional outcomes. Although the reasons for such a poor prognosis remain unanswered, this category of individuals possibly display a gap in their immune repertoire (against VZV), and as such, deserve specific medical attention.

There are limitations that need to be acknowledged. They involve the retrospective nature of data collection precluding sample size calculation and selection bias of patients referred to a tertiary center. Reported results are based on small samples; however, this cohort remains one of the largest ever published on this subject. We included CMV retinitis alongside HSV and VZV VRN because the former is an interesting comparator to the latter. We chose two arbitrary surrogates to define a poor outcome, namely an anatomic one: the occurrence of RD; and a functional one: BCVA ≤ 20/100 (i.e., ≥0.7 LogMAR) at month 1. The two surrogates were studied separately in terms of mathematics. It is agreed that some patients may experience RD and still retain a good BCVA. However, the disease burden increases significantly with RD because it implies surgery. The choice of the two surrogates accounts for this additional burden and seems to be clinically relevant. The definition of rapid and slow responders can be questioned as it depends on different endpoints across included patients. This is inherent to the retrospective nature of the study. However, Appendix A shows that the median endpoints used for slow and rapid responder definition ranged between 13 and 14.5 days, which is comparable between included groups, independent of the causative virus or immune status (*p* = 0.82). None of the patients received systemic steroids. Local steroids were used in all patients at doses ranging from 3 to 6 drops/day with progressive tapering, which explains the absence of correlation between steroid use and presented outcomes (data not shown). Finally, we acknowledge that important predictive factors of poor visual acuity or RD occurrence (e.g., degree of myopia, pre-existing retinal lesions, rate of pseudophakic eyes, presence or absence of optic nerve atrophy or retinal vasculitis) were not evaluated. Future prospective studies will be of paramount importance to look at their involvement.

## 5. Conclusions

In conclusion, our study supports previously published predictive factors of poor visual evolution, namely, older age, low baseline visual acuity, large necrosis areas and high vitritis grade [9]. The newest aspect of this work was the analysis of virological parameters. We showed that high baseline VL and the “slow responder” status were both associated with poor outcomes independently of the causative virus or immune status. Immunocompetent VZV-infected patients deserve specific attention as they show the worst prognosis. These new data should help physicians with their daily management of VRN patients.

## Figures and Tables

**Figure 1 jpm-12-01785-f001:**
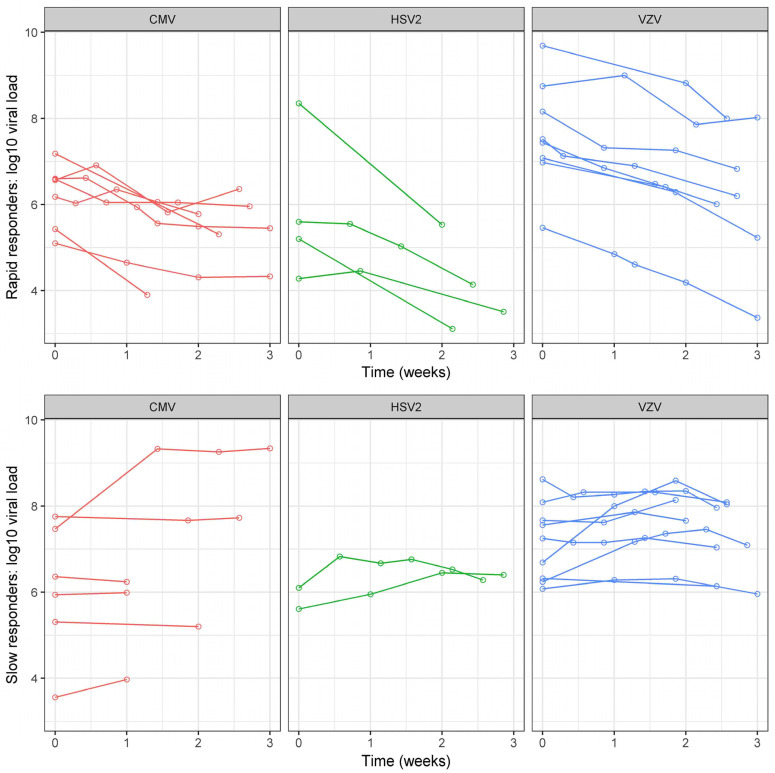
Viral load evolution in rapid and slow responders according to each virus. Each virus is presented with a different color. Each eye is represented by a single curve. Slow and rapid responders correspond respectively to eyes with a viral load decrease lower or greater than 50% (versus baseline) after 2 +/− 1 weeks of intravenous antiviral treatment. On the X axis, viral load measured at the 1st visit (baseline = 0) and after 1, 2 and 3 weeks. On the Y-axis, viral load expressed in the logarithm of international units per milliliter. HSV: Herpes simplex virus; VZV: Varicella zoster virus; CMV: Cytomegalovirus.

**Figure 2 jpm-12-01785-f002:**
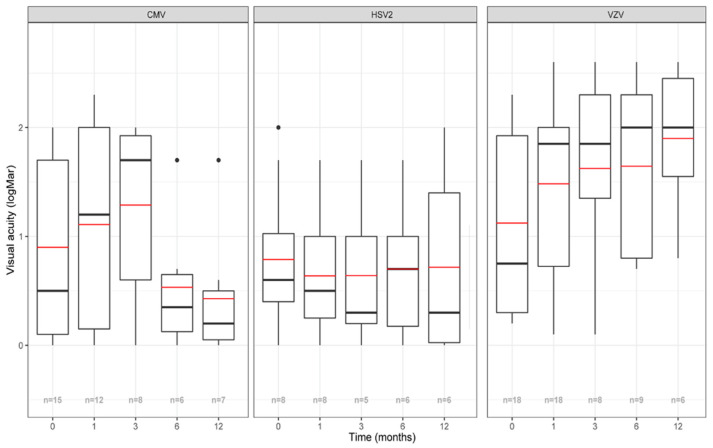
Evolution of visual acuity according to each virus. X-axis: time in months. Y-axis: BCVA (best-corrected visual acuity) in LogMAR. The red line represents the mean BCVA, the black line represents the median BCVA, the n displayed at the bottom corresponds to the number of evaluated eyes at each timepoint. Single points represent an extreme value < Q1-1.5*Interquartile range or > Q3+1.5*Interquartile range. HSV: Herpes simplex virus; VZV: Varicella zoster virus; CMV: Cytomegalovirus.

**Table 1 jpm-12-01785-t001:** Patients and eye characteristics at baseline according to the causal virus.

	Virus				
Variables (Described in Terms of Patients)	CMV (N = 13)	HSV2 (N = 8)	VZV (N = 16)	Total (N = 37)	*p* Value
Gender (Females), N (%)	3/13 (23.1%)	3/8 (37.5%)	8/16 (50%)	14/37 (37.8%)	0.36
Age, Mean (SD)	59.6 (17.9)	37.9 (13.1)	63.8 (14.9)	56.7 (18.4)	0.005
Immunodepression, N (%)	13/13 (100%)	0/8 (0%)	4/16 (25%)	17/37 (45.9%)	<0.001
**Variables (described in terms of eyes)**	**CMV** **(N = 15)**	**HSV2** **(N = 8)**	**VZV** **(N = 18)**	**Total** **(N = 41)**	
Baseline BCVA (LogMAR), Mean (SD)	0.9 (0.8)	0.8 (0.7)	1.1 (0.8)	0.9 (0.8)	0.37
Baseline vitritis grade, Mean (SD)	1.6 (1.5)	2.3 (0.7)	2.4 (0.9)	2.1 (1.1)	0.15
Number of retinal quadrants affected by retinal necrosis, Mean (SD)	1.9 (1.2)	1.4 (1.0)	2.6 (1.4)	2.1 (1.3)	0.11
Viral load at baseline (Log10), Mean (SD)	6.2 (1.1)	5.3 (1.5)	7.4 (1.0)	6.6 (1.4)	0.001

HSV: herpes simplex virus; VZV: varicella zoster virus; CMV: cytomegalovirus; N: number; %: percentage; SD: standard deviation.

**Table 2 jpm-12-01785-t002:** Clinical and virological characteristics of eyes by virus type during the follow-up.

	Virus				
Variables (Described in Terms of Eyes)	CMV (N = 15)	HSV2 (N = 8)	VZV (N = 18)	Total (N = 41)	*p*-Value
Bilateralization of viral necrotizing retinitis, N patients (%)	5/13 (38.5%)	3/8 (37.5%)	2/16 (12.5%)	10/37 (27.0%)	0.25
Time to bilateralization from 1st visit (months), Mean (SD)	11 (20.9)	88 (13.9)	0.5 (0.7)	32 (41.8)	0.046
Retinal detachment during follow-up, N (%)	3/15 (20%)	2/8 (25%)	6/18 (33.3%)	11/41 (26.8%)	0.82
Time from diagnosis to RD (days), Mean (SD)	119 (183.2)	60 (84.8)	101.7 (122.4)	98.8 (123.9)	0.80
Time from symptom onset to treatment (days), Mean (SD)	54.2 (73.5)	4.9 (4.9)	10.2 (14.2)	21.7 (44.4)	0.001
Duration of intravenous treatment (days), Mean (SD)	29 (29.2)	21	20.4 (9.8)	23.5 (18.2)	0.79
IV Acyclovir, N (%)	4/12 (33.3%)	4/8 (50%)	14/18 (77.8%)	22/38 (57.9%)	0.04
IV Ganciclovir, N (%)	8/13 (61.5%)	0/8 (0%)	0/18 (0%)	8/39 (20.5%)	0.0005
IV Foscarnet, N (%)	6/13 (46.2%)	5/8 (62.5%)	13/18 (72.2%)	24/39 (61.5%)	0.35
Number of Ganciclovir IVI, Mean (SD)	9.2 (10.9)	4.6 (2.9)	5.8 (5.7)	6.8 (7.7)	0.59
Number of Foscarnet IVIs, Mean (SD)	1.7 (4.5)	1.6 (4.6)	5.7 (8.1)	3.4 (6.5)	0.10
Viral load response at 2 +/− 1 weeks of intravenous antiviral therapy initiation (rapid responders), N (%)	7/13 (53.8%)	4/6 (66.7%)	8/17 (47.1%)	19/36 (52.8%)	0.84

HSV: herpes simplex virus; VZV: varicella zoster virus; CMV: cytomegalovirus; BCVA: best corrected visual acuity; N: number; %: percentage; IVI: intravitreal injection; IV: intravenous; RD: retinal detachment; SD: standard deviation. Rapid responder: 50% or greater decrease of the baseline viral load after 2 +/− 1 weeks of intravenous antiviral treatment.

**Table 3 jpm-12-01785-t003:** Univariate analysis of factors associated with BCVA ≤ 20/100 at Month 1.

	BCVA ≤ 20/100 at Month 1			
Variable	No (N = 14)	Yes (N = 24)	Total (N = 38)	*p*-Value
Age, Mean (SD)	47.3 (14.7)	63.0 (18.8)	58.1 (18.6)	0.015
Females, N (%)	4/14 (28.6%)	13/24 (54.2%)	17/38 (44.7%)	0.12
Immunodepression, N (%)	8/13 (61.5%)	9/24 (37.5%)	17/37 (45.9%)	0.16
Virus				0.14
CMV	5/14 (35.7%)	7/24 (29.2%)	12/38 (31.6%)	
HSV2	5/14 (35.7%)	3/24 (12.5%)	8/38 (21.1%)	
VZV	4/14 (28.6%)	14/24 (58.3%)	18/38 (47.4%)	
Baseline BCVA (LogMAR), Mean (SD)	0.6 (0.6)	1.3 (0.8)	0.9 (0.8)	0.002
Baseline vitritis (grade +), Mean (SD)	1.7 (0.9)	2.4 (1.1)	2.1 (1.1)	0.08
Number of retinal quadrants with necrosis at baseline, Mean (SD)	1.6 (1.1)	2.5 (1.4)	2.1 (1.3)	0.07
Viral load (Log 10) at baseline, Mean (SD)	5.7 (1.4)	7.1 (1.2)	6.6 (1.4)	0.007
Time from symptom onset to initiation of treatment (days), Mean (SD)	15.2 (33.4)	14.1 (15.2)	21.7 (44.4)	0.11
IV Acyclovir, N (%)	6/14 (42.9%)	15/21 (71.4%)	21/35 (60%)	0.09
IV Ganciclovir, N (%)	2/14 (14.3%)	4/22 (18.2%)	6/36 (16.7%)	1
IV Foscarnet, N (%)	9/14 (64.3%)	14/22 (63.6%)	23/36 (63.9%)	0.96
Total number of IVIs, Mean (SD)	10.5 (10.2)	10.7 (7.2)	10.2 (8.2)	0.54
Number of Ganciclovir IVIs, Mean (SD)	8.9 (10.6)	5.9 (5.9)	6.8 (7.7)	0.29
Number of Foscarnet IVIs, Mean (SD)	1.6 (4.2)	4.9 (7.7)	3.4 (6.5)	0.13
Rapid responder, N (%)	8/11 (72.7%)	8/22 (36.4%)	16/33 (48.5%)	0.04

HSV: herpes simplex virus; VZV: varicella zoster virus; CMV: cytomegalovirus; BCVA: best corrected visual acuity; N: number; %: percentage; IVI: intravitreal injection; IV: intravenous; SD: standard deviation; M1: month 1; rapid responder: 50% or greater decrease of the baseline viral load after 2 +/− 1 weeks of intravenous (induction) antiviral treatment.

**Table 4 jpm-12-01785-t004:** Logistic analysis of the predictive factors of BCVA ≤ 20/100 at one month of treatment.

Univariate Analysis				
Variable	N (Eyes)	OR	CI 95%	*p*-Value
Virus HSV2 (ref = CMV)	38	0.43	0.07–2.68	0.131
Virus VZV (ref = CMV)	38	2.5	0.51–12.35	
Age (years)	38	1.29	1.04–1.6	0.01
Baseline BCVA (LogMAR, ^x^10)	38	1.15	1.03–1.28	0.005
Vitritis (+)	36	1.96	1.01–3.98	0.047
Baseline viral load (Log10)	38	2.45	1.24–4.84	0.002
Slow responder (yes)	33	4.67	1.01–22.79	0.046
**Multivariate analysis adjusted on virus type ^a^**				
Baseline viral load (Log10)	38	2.32	1.04–5.17	0.018
Slow responder (yes)	33	4.91	1.01–26.04	0.049

BCVA: best corrected visual acuity; OR: odds ratio; CI: confidence interval; Ref: reference; slow responder: decrease in viral load of less than 50% after 2 +/− 1 weeks of induction antiviral therapy. ^a^ Multivariate model integrating the type of virus.

**Table 5 jpm-12-01785-t005:** Factors associated with the occurrence of retinal detachment during the follow-up.

	Occurrence of Retinal Detachment			
Variable	No (N = 30)	Yes (N = 11)	Total (N = 41)	*p*-Value
Age, Mean (SD)	54.8 (16.9)	66.8 (20.9)	58.0 (18.6)	0.07
Females, N (%)	13/30 (43.3%)	4/11 (36.4%)	17/41 (41.5%)	0.73
Immunodepression, N (%)	17/29 (58.6%)	3/11 (27.3%)	20/40 (50%)	0.07
Virus, N (%)				0.82
CMV	12/30 (40%)	3/11 (27.3%)	15/41 (36.6%)	
HSV2	6/30 (20%)	2/11 (18.2%)	8/41 (19.5%)	
VZV	12/30 (40%)	6/11 (54.5%)	18/41 (43.9%)	
Baseline BCVA (LogMAR), Mean (SD)	0.7 (0.7)	1.6 (0.7)	0.9 (0.8)	0.003
Baseline vitritis (+), Mean (SD)	1.9 (1.0)	2.7 (1.2)	2.1 (1.1)	0.02
Number of retinal quadrants with necrosis at baseline, Mean (SD)	1.9 (1.2)	2.6 (1.4)	2.1 (1.3)	0.26
Viral load (Log 10) at baseline, Mean (SD)	6.3 (1.3)	7.2 (1.4)	6.6 (1.4)	0.14
Time from symptom onset to initiation of treatment (days), Mean (SD)	24.2 (49.5)	13.2 (19.0)	21.7 (44.4)	0.97
IV Acyclovir, N (%)	16/28 (57.1%)	6/10 (60%)	22/38 (57.9%)	1
IV Ganciclovir, N (%)	8/29 (27.6%)	0/10 (0%)	8/39 (20.5%)	0.08
IV Foscarnet, N (%)	17/29 (58.6%)	7/10 (70%)	24/39 (61.5%)	0.71
Number of IVIs (total), Mean (SD)	9.4 (5.3)	12.4 (13.4)	10.2 (8.2)	0.80
Number of IVI ≤ 5 N (%)	13 (43.3%)	9 (81.8%)	22 (53.7%)	0.02
Rapid responder, N (%)	14/25 (56%)	5/11 (45.5%)	19/36 (52.8%)	0.55

HSV: herpes simplex virus; VZV: varicella zoster virus; CMV: cytomegalovirus; BCVA: best corrected visual acuity; N: number; %: percentage; IVI: intravitreal injection; IV: intravenous; SD: standard deviation; Rapid responder: 50% or greater decrease of the baseline viral load after 2 +/− 1 weeks of induction treatment.

## Data Availability

Not applicable.

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
