# Peer review of "Clinical and Virological Characteristics and Prognostic Factors in Viral Necrotizing Retinitis"

_jpm, 2022, doi:10.3390/jpm12111785_

Round 1
Reviewer 1 Report
This is a very interesting study about an unfrequent and poorly recognized viral retinal disease with dramatic consequences if left untreated. I have no specific comments pertaining to this manuscript.
Reviewer 2 Report
However, the presented topic is interesting and important for ophthalmologists, the paper was not presented in a reader friendly way. In some paragraphs, the authors did not put references; tables are very long and hard to understand; the subjects did not characterize enough etc.; sample size calculation is needed. The discussion is poor.
In addition, the authors should have read the Journal's requirements before submit the paper.
Reviewer 3 Report
It would be nice to :
State if the clinical characteristics of slow and rapid responders differ.
State if a low dose of systemic steroid has been given and if it did influence the viral load.
Discuss on the fact that immunocompetent VZV group have a worse prognosis than immunosuppressed VZV group (?more aggressive treatment ?lower RD occurrence because of lesser vitritis) & on the 0% HSV1 (in comparison to the literature).
Round 2
Reviewer 2 Report
It has to be recognized the critical effort made by the authors to improve the article. However, I still consider that some aspects justify not recommending its publication in the current state.
I agree with the authors that their paper is retrospective, but
a power calculation should have been done if this was a retrospective study.